# Conformational Dynamics of Mitochondrial Inorganic Pyrophosphatase hPPA2 and Its Changes Caused by Pathogenic Mutations

**DOI:** 10.3390/life15010100

**Published:** 2025-01-15

**Authors:** Ekaterina Bezpalaya, Svetlana Kurilova, Nataliya Vorobyeva, Elena Rodina

**Affiliations:** 1Chemistry Department, Lomonosov Moscow State University, 119991 Moscow, Russia; bezpalaya.katya@gmail.com; 2Belozersky Institute of Physico-Chemical Biology, Lomonosov Moscow State University, 119899 Moscow, Russia; kurilova@belozersky.msu.ru (S.K.); nvorob@yandex.ru (N.V.)

**Keywords:** inorganic pyrophosphatase, mitochondrial, molecular dynamics simulation, conformational dynamics, principal component analysis, normal mode analysis

## Abstract

Inorganic pyrophosphatases, or PPases, are ubiquitous enzymes whose activity is necessary for a large number of biosynthetic reactions. The catalytic function of PPases is dependent on certain conformational changes that have been previously characterized based on the comparison of the crystal structures of various complexes. The current work describes the conformational dynamics of a structural model of human mitochondrial pyrophosphatase hPPA2 using molecular dynamics simulation, all-atom principal component analysis, and coarse-grained normal mode analysis. In addition to the wild-type enzyme, four mutant variants of hPPA2 were characterized that correspond to the natural pathogenic variants causing severe mitochondrial dysfunction and cardio pathologies. As a result, we identified the global type of flexible motion that seems to be shared by other dimeric PPases. This motion is discussed in terms of the allosteric behavior of the protein. Analysis of the observed conformational dynamics revealed the formation of a binding site for anionic ligands in the active site that could be relevant to enzyme catalysis. Based on the comparison of the wild-type and mutant PPases dynamics, we suggest the possible molecular mechanisms of the functional incompetence of hPPA2 caused by mutations. The results of this work allow for deeper insight into the structural basis of PPase function and the possible effects of pathogenic mutations on the protein structure and function.

## 1. Introduction

This work is aimed at the characterization of a human mitochondrial pyrophosphatase hPPA2 and the effects of mutations on its function.

Inorganic pyrophosphatases (PPases, E.c. 3.6.1.1) catalyze metal-dependent hydrolysis of pyrophosphate PP_i_ to phosphate P_i_. Soluble PPases (sPPases) are grouped into non-homologous structural Families I and II. In addition to sPPases, there are integral membrane enzymes of separate groups (mPPases) that can couple PP_i_ hydrolysis with H^+^ or Na^+^ translocation through biological membranes. Soluble PPases are important housekeeping enzymes; their most known metabolic role is the thermodynamic pull of endergonic biosynthetic reactions that yield PP_i_ as a by-product [1]. Among sPPases, Family I PPases are the most widely spread enzymes found in all domains of life. In animals, Family I PPases are solely responsible for PPase function. The human nuclear genome contains two genes encoding two different Family I PPases, the cytosolic enzyme hPPA1 [2] and the mitochondrial enzyme hPPA2 [3]. These two proteins are highly similar, and their amino acid sequences are 60% identical. Due to the significance of PPase activity for the cell, its disruption can cause severe metabolic consequences. hPPA1 is overly expressed in cancer cells where it promotes tumor development through several signaling pathways and is considered a negative prognostic biomarker in various types of cancer [4,5,6]. hPPA2 is expressed at a much lower level than hPPA1, and its metabolic role is not yet clearly understood. Functional studies of PPA2 in yeast *Saccharomyces cerevisiae* suggested its importance for the mitochondrial metabolism and maintenance of mitochondrial DNA [7]. In humans, biallelic mutations in the gene *ppa2* impair both respiratory and other metabolic functions of mitochondria, which in most patients leads to early death due to sudden cardiac arrest [8,9,10]. Several studied mutations of hPPA2 were found to decrease catalytic activity, thermostability, and/or protein expression levels [8,9,10]. A deficiency of total PPase activity in the mitochondrial matrix caused by either of these reasons is believed to be the primary cause of mitochondrial dysfunction. Among 11 missense variants classified in the ClinVar database as pathogenic or likely pathogenic, most mutated residues have no obvious connection to the active site or other functionally important sites of PPA2. Therefore, the exact molecular bases of the pathogenic effects of specific mutations on hPPA2 function are still unclear.

In our recent study, we solved the crystal structure of the PPA2 enzyme from thermotolerant yeast *Ogataea parapolymorpha* (OpPPA2) and characterized its mutant variant Met52Val corresponding to the pathogenic variant Met94Val of the human hPPA2 [11]. This mutant variant retained the overall structure and stability but demonstrated critically impaired catalytic properties compared to the wild-type PPase. Based on the obtained data, we suggested that the reason for the observed effects was the impaired conformational flexibility of mutated PPA2 which, in particular, affected the catalytically important structural transitions.

Numerous experimental data demonstrate that the conformational changes are key for the catalytic activity of sPPases [12,13,14]. For the two Family I cytosolic PPases, those from *Escherichia coli* (Ec-PPase) and *Saccharomyces cerevisiae* (ScPPA1), the available crystal structures made it possible to describe the conformational changes along the catalytic pathway [12,13,14]. The scheme of catalysis by ScPPA1 includes six catalytic intermediates A–F starting from the holoform where PPase is complexed with cofactor metal ions M1 and M2 [12]. In the intermediates A–F, the enzyme can adopt either “open” or “closed” conformation manifested in particular by the position of the active site loops I–III [12]. “Open” conformation is found in the holoform, while the following binding of substrate MgPP_i_ is accompanied by the transition to the “closed” status, the shift of M1 and M2 closer to each other, and the formation of an attacking nucleophile [12]. The crucial importance of the inherent flexibility of the active site loops was demonstrated by the mutational analysis in Ec-PPase: mutations of the conservative Gly residues in loops II and III slowed down the equilibration with cofactor Mg^2+^ and decreased the catalytic efficiency by two orders of magnitude [13].

In addition to being part of the catalysis, conformational dynamics is believed to be an important part of the allosteric behavior of PPases. The heterotropic effects of allosteric ligands on the functional properties were demonstrated for various sPPases [15,16,17,18,19]. Binding sites for allosteric effectors were found, in particular, in the intersubunit interfaces [19,20]. In the crystal structures of oligomeric sPPases, subunit asymmetry was observed; subunits differed in the small-molecule ligand content, cofactor metal ions coordination, and the orientation of the active site loops (see, e.g., [12,21]). The observed differences may result from the specific homotropic interaction between the active sites; alternatively, they may reflect the random conformational changes of flexible structural elements or be shaped by crystal contacts. The possible significance of subunit asymmetry for sPPase function is still unclear. The structural bases of homo- or heterotropic allosteric regulation of sPPases are poorly understood.

Therefore, our current research was motivated by the idea that the point mutations in hPPA2, without significant effects on the protein structure or stability, can disrupt the conformational transitions important for the PPase function, which can be the major reason for their pathogenicity. In order to test this hypothesis, in this work we studied the conformational dynamics of hPPA2, using molecular dynamic simulation of its 3D structural model, and of hPPA1 using its crystal structure. Obtained data were analyzed by the principal component analysis, normal mode analysis and other approaches to visualize structural changes of the protein molecule in a solution. This analysis was performed on the wild-type hPPA2 as well as on the structural models of four mutant variants with point mutations corresponding to the pathogenic variants of hPPA2. Based on this analysis, we describe the global conformational dynamics of hPPA2 and homologous PPases. We reveal the novel catalytically important conformational substates within the PPase holoform manifested in the different subunits of dimeric hPPA2. We identify the conformational changes impaired by the studied mutations and discuss their possible functional implications. The results of this work allow for a deeper insight into the structural basis of PPase function and the possible effects of pathogenic mutations on the protein structure and function. Our results will be useful in future research on the allosteric behavior of PPases and possible mechanisms of their regulation and protein–protein interactions.

## 2. Materials and Methods

Evaluation of mutation effects on fold stability: ΔΔG of hPPA2 were calculated using a sequence-based variant of the Strum algorithm (https://zhanggroup.org/STRUM/about.html (accessed on 4 May 2023) [22]). Conservation scores were calculated using ConSurf server (https://consurf.tau.ac.il/consurf_index.php (accessed on 14 February 2023) [23,24]).

Model Preparation: a 3D structure of hPPA2 was modeled using I-Tasser server (https://zhanggroup.org/I-TASSER/ (accessed on 19 November 2021) [25]) based on the Uniprot (https://www.uniprot.org/ (accessed on 17 October 2021), [26]) sequence Q9H2U2 (IPYR2_HUMAN). The mature form of a protein (residues 33–334) without an N-terminal mitochondrial transfer peptide was used for modelling. The 5 top-ranked models produced by I-Tasser were analyzed for reliability using local quality criteria. The best-ranked model had a C-score of 0.44 and TM-score of 0.66, and hPPA1 (PDB ID 6C45 [27]) was recognized as the best template. All models had the same protein fold and slightly differed only in the conformations of highly flexible chain fragments, which were modeled with the lowest confidence score (residues 71–90 and 242–249). All generated models had good scores in validation estimated by the percentage of Ramachandran favored and unfavored residues, the MolProbity score, and the Clash Score and were of high reliability. The best-ranked model was used in further analysis. The obtained model of the hPPA2 subunit was subsequently used to manually model dimeric hPPA2 in a holoform. For the holoform modeling, Mg^2+^ at the sites M1 and M2 and coordination water molecules were adopted from the structure of the holoform of yeast PPase 1WGI [28]. Dimer was generated by symmetry modeling using the crystal structure of human cytoplasmic PPase hPPA1 (6C45 [27]), from which crystallographic water molecules were also included. The hPPA2 variants (Ser61Phe, Met94Val, Met106Ile, Gln294Pro) were created by substituting respective residues for mutated ones in both subunits of the WT hPPA2 structure using UCSF Chimera (version 1.16) [29]. Model construction and validation were completed in WinCoot (version 1.1.11) [30]. All models were corrected manually by removing steric clashes with a 0.6 Å van der Waals overlap threshold, adjusting amino acid rotamers, and correcting peptide bond angles to reduce Ramachandran outliers to five or fewer. Protonation states were assigned to the residues at pH 7.5 according to p*K*_a_ values calculated using ProPka server (https://www.ddl.unimi.it/vegaol/propka.htm, accessed on 22 November 2021) [31]. Hydrogen atoms were added to the models using the H++ server (http://newbiophysics.cs.vt.edu/H++/ (accessed on 25 November 2021) [32]). Active site aspartate side chains were deprotonated manually if necessary. Molecular visualization and interactive model adjustments were performed in UCSF Chimera and VMD (version 1.9.2) [33].

System Preparation and Molecular Dynamics Simulation: Molecular dynamics simulations were carried out using AmberTools22 (version for Ubuntu 20.4) [34]. Each protein model was placed within a truncated octahedral water box. Solvent shells were placed around the protein models using placevent. K^+^ and Cl^−^ ions were added to mimic 0.15 M KCl solution. The final solvent density estimated by 3D-RISM was of 55.5 mol/L. Amber was used for structure parametrization. Calculations were performed on a graphical station with 4 NVIDIA GeForce GTX Titan X graphics cards, operated under Ubuntu 14 with a Cuda toolkit package (version 6.5). The prepared systems underwent simulations at 300 K and constant pressure. The temperature was kept constant using a Langevin thermostat, and the pressure (1.0 bar) was kept constant using a Berendsen barostat. System relaxation involved two-stage minimization in sander, first with constraints on protein atoms to relax solvent, then fully minimizing the system. Heating from 0 K to 300 K was achieved in 10,000 steps with a 2 ps time step. Production simulations were then conducted subsequently for 50 ns using pmemd.cuda_SPFP, with non-bonded interactions calculated by a 10 Å cutoff. Each system was simulated twice for a cumulative 200 ns (WT hPPA2 and Met94Val) or 100 ns (rest of the proteins) per setup. Models were independently reassembled for each dynamics repeat.

Trajectory Analysis: A trajectory analysis was performed using CPPTRAJ (version 6.4.4), VMD and Python’s (version 3.10) correlationplus. Data processing, visualization and analysis was performed using Python libraries, including numpy, pandas, matplotlib, seaborn, scipy, correlationplus, and placevent. Figures were made using UCSF Chimera, VMD and SigmaPlot (version 11.0).

NMA analysis: Coarse-grained Normal Mode Analysis (NMA) was performed using portal DynOmics http://dynomics.pitt.edu/ (accessed on 04.11.2024) [35]. This portal utilizes an anisotropic network model (ANM) [36] for molecular motions and animations and a Gaussian network model (GNM) [37] for the analysis of RMSF, cross-correlations between residue fluctuations, an inter-residue contact map, and properties of the GNM mode spectrum. Data were visualized using VMD or downloaded as the figures from the server.

## 3. Results

### 3.1. Analysis of the Possible Effects of Pathogenic Mutations on hPPA2 Structure

In order to characterize the possible impact of pathogenic mutations on protein structural stability, we calculated the conservation scores and predicted changes in folding free energy, ΔΔG (Appendix A). The location of the mutated residues in the hPPA2 structure and their conservation scores are illustrated in Figure 1. Most of the mutated residues are graded 8 or 9 by the ConSurf algorithm, and their conservation scores calculated by Strum are negative, which corresponds to highly conserved positions. Obtained data suggest that mutations of Met94, Glu172 and Trp202 to any residue should be destabilizing since aggregated ΔΔG for these residues have moderate but clearly negative values. The rest of the pathogenic mutations show only a slight, if any, destabilizing effect with positive or close to zero values of ΔΔG. Therefore, the destabilization of a protein structure itself is probably not the primary cause of the pathogenic effect of mutations. However, the effect of mutations on the protein structure could be revealed after the analysis of dynamic rather than static conformational states. That prompted us to study the conformational dynamics of hPPA2 and identify the changes in the dynamic behavior of the protein caused by mutations.

### 3.2. Conformational Dynamics of hPPA2 Studied by Molecular Dynamics Simulation

#### 3.2.1. General Characteristics

The 3D structure of hPPA2 is still unknown; however, several crystal structures of eukaryotic Family I PPases are available in pdb and they all share a highly conserved global fold and dimeric organization. The structure of hPPA2 was, therefore, homology modeled in a holoform, optimized and prepared for the molecular dynamic simulations, as described in the Materials and Methods. After minimization and heating to 300 K, two runs of molecular dynamic simulations were performed for 200 ns each. RMSD profiles for the wild-type hPPA2 are shown in Appendix A. The histograms of RMSD for the two subunits of a dimer are slightly different (Appendix A, bottom plot); this result demonstrates that, although the original dimer of hPPA2 was generated from the identical subunits by their symmetry-related duplication, MD simulation induced subunit asymmetry in hPPA2. Superposition of the two subunits of dimers from the random frames along the trajectory gives an RMSD of 2.1–2.4 Å. The major variations between subunits (shown in Appendix A) involve the regions of a chain 71–90 forming the Ω-loop, 121–128 (active site loop I), 149–166 (active site loop II), and 290–306. The observed differences of these regions remain similar for several frames of a trajectory, demonstrating that they form a specific pattern rather than random fluctuations. These parts of a chain are not directly involved in the intersubunit contact, so these differences may result from the information transfer between subunits and hence may be relevant to the allosteric properties of PPase.

General flexibility of hPPA2 as analyzed by the all-atom RMSF analysis of hPPA2 trajectory after its equilibration is illustrated in Figure 2 and Appendix A. The most flexible chain fragments fluctuating with an amplitude of 2.5 Å and higher include several chain regions colored red in Figure 2. The Ω-loop 71–90 is unique for the structure of mitochondrial PPases from animals; it is located at the protein surface and is not stabilized by the contacts with the main body of a protein. The functional role of this loop is still unknown. The coil regions 121–127 and 152–161 (active site loops I and II) contain the residues important for the catalysis and determine the “open” or “closed” state of the active site. The extended region 295–320 is located close to the C-terminus of hPPA2 and does not contain known residues of functional importance. As expected, the most stable and least fluctuating residues (colored blue in Figure 2) belong to the constituents of a β-barrel and elements stabilized by an intersubunit interaction. As in the homologous Family I PPases with known crystal structures, most of the active site residues belong to the stable β-strands involved in barrel formation; however, some active site residues (Arg127, Asp164, Asp201, Lys203) belong to less ordered structural elements and demonstrate moderate flexibility.

As demonstrated in Appendix A where the RMSF calculated for the different parts of an MD trajectory are shown, the RMSF profile only changes at the beginning of the simulation, whereas after approximately the 100th frame, the profiles are qualitatively similar.

#### 3.2.2. Metal Coordination

Cofactor Mg^2+^ ions modeled at M1 and M2 sites are retained in both subunits for the entire course of the simulation. The protein ligands of Mg^2+^ in subunit A are carboxylic groups of Asp164, Asp169 and Asp201 as it was modeled in the original structure (Figure 3A). The side chains of Glu97, Glu107, Asp166 and Tyr142 as well as the peptide oxygen of Pro167 coordinate metal ions via water molecules. In subunit B, carboxylic oxygen atoms of Asp201 are approximately 4 Å from the metal ion and, instead, the fourth water molecule is coordinated to Mg^2+^, while the other protein ligands are the same (Figure 3B, Appendix A). Fluctuations of both metal ions around their mean positions are in the range of 0.6–0.7 Å in both subunits (Appendix A). As described in the Introduction, while in the holoform, the metal ions M1 and M2 are connected by a two-water bridge, subsequent substrate binding brings the metal ions closer to each other and one of these two water molecules becomes a bridging ligand and an attacking nucleophile [12]. In the course of the MD simulation, metal ions in both subunits do not come closer than 5.2 Å to each other (Appendix A), and they stay connected by a two-water bridge as in the original structure. In some frames, Cl^-^ is bound in the vicinity of Mg^2+^; however, this is not enough to bring metal ions closer. Presumably, the formation of a one-water bridge requires the binding of a larger anion (e.g., phosphate or pyrophosphate) that can coordinate both metal ions simultaneously and thus stimulate their movement to each other. In subunit B of dimeric hPPA2, due to the alternative positions of the regions II and III containing metal ligands Asp164 and Asp201, the concomitant change in M1 coordination is observed (Figure 3B). This difference between the subunits may imply that the two subunits coordinate their dynamics in the course of catalysis.

#### 3.2.3. Anionic Binding Site

In the preparation of the protein model for the MD simulation, K^+^ and Cl^−^ were used as counter-ions to mimic the physiological conditions. The analysis of the distribution of Cl^-^ in the simulation box demonstrates that they form a cluster within the protein active site where the probability of finding Cl^−^ is significantly higher than at any other place in the box. In contrast, K^+^ shows an even probability distribution without clustering. Figure 4 shows a high-probability cluster of Cl^−^ ions in the active site in the vicinity of Mg^2+^ ion M2. Such clustering of Cl^−^ in one place suggests the formation of a binding site for anionic ligands. The position of this cluster coincides with the position of pyrophosphate in the enzyme–substrate complex of a homologous PPase, as observed in the crystal structure 2AUU ([32], Figure 4B). We may propose, therefore, that the conformational dynamics of a holo-form of hPPA2 in a solution leads to the formation of a binding site, which can be occupied by pyrophosphate as part of a substrate.

It should be noted that this binding site manifested by Cl^-^ clustering was observed only in one subunit of a dimeric molecule (subunit B). Analogous clusters of Cl^-^ were found in some of the mutant variants of hPPA2 (see below). This observation, therefore, follows some pattern, though the structural basis for that remains to be understood.

#### 3.2.4. Conformation of the Active Site Loops

As was mentioned above, the active site loops I–III define the “open” or “closed” status of the active site, which changes along the catalytic cycle [12]. Figure 5 shows the representative conformation of hPPA2 after MD simulation superimposed onto the “open” and “closed” conformations as they were observed for the Sc-PPase (PDB ID: 2IHP, subunit A is “open” and subunit B is “closed” [12]). The loops determining “open” or “closed” state are shown in Sc-PPase by blue and cyan colors, respectively, and in hPPA2, the corresponding regions of a chain are highlighted in purple color. These regions of a chain contain the residues important for metal binding (Asp201, Asp196), pyrophosphate or phosphate coordination (Arg127), or contribute the release of phosphate P1 (Lys122 and Lys125).

As expected for the holoform, the conformation of hPPA2 represents the “open” state, which is even more pronounced in subunit B. This state is maintained, in particular, by the interactions of the residues mentioned above with other parts of a molecule shown in Figure 5. For instance, in the absence of M3, Asp196 from loop II interacts with Asn250 and Lys242; in the absence of PP_i_, Arg127 from loop I interacts with Asp245, and Lys125 with Glu325. These interactions populate a large fraction of frames in the course of MD simulation. We may expect that the substrate binding would trigger the partner exchange in these residues and the concomitant “closure” of the active site. In particular, the motion of a loop II containing Asp164 has probably the largest impact on the motion of M1 required for the one-bridging water state of the active site and the formation of an attacking nucleophile.

#### 3.2.5. PCA Analysis of MD Trajectory

The conformational space of hPPA2 in the course of MD simulation was characterized using all-atom Principal Component Analysis (PCA). This approach allows to describe the most essential (in terms of amplitude) motions of the molecule, where calculated parameter PC1 (principal component 1) corresponds to the greatest variance, PC2—to the next greatest variance, etc. Different conformational states are characterized by different values of PC1 and PC2, and the plot of PC2 against PC1 illustrates the relations between conformations available for the protein. Figure 6 shows the possible conformational space of WT hPPA2 calculated from the MD trajectory. The observed conformations can be grouped into two clusters with population density higher than average (clusters I and II in Figure 6A, green or yellow symbols), and in addition, there is a lot of separate states not forming dense clusters (blue or purple symbols). PCA analysis was performed using the entire simulation time; in order to visualize the transitions between conformations in the course of the simulation, the same data are presented in Figure 6B with the color scheme corresponding to a frame of the trajectory: from the beginning 100 frames (blue symbols) to the final 100 frames (purple symbols). This analysis suggests that in the course of MD simulation, the constant drift is observed from one group of conformations to another group, then at some point in the middle of the simulation, the structure stays at a more populated cluster II, and finally, it comes to the most populated cluster I and does not drift further. The cluster analysis of the trajectory using K-means or Dbscan approaches is in good agreement with these results.

While the analysis of RMSF identifies the most flexible parts of a molecule in general, it does not reveal if these parts move in concert or independently, and it cannot distinguish between directed or random movements. The PCA analysis of the protein conformational space reveals the direction of the observed movements. Figure 7A shows which parts of an hPPA2 molecule contribute most to a PC1, i.e., the highest-amplitude motion mode. According to these calculations, PC1 involves several regions of a protein including the Ω-loop 71–90, flexible regions I (121–128) and II (155–161) around the active site, and other protein parts labeled in Figure 7. Directions of their oscillation are visualized on a porcupine plot (Figure 7B) by purple arrows. The predominant character of conformational changes in PC1 is a rocking motion of subunits causing the shift of these atoms in the observed directions.

#### 3.2.6. NMA Analysis of hPPA2 Conformational Dynamics

While PCA analysis classifies observed motions based on their amplitudes, another widely used approach, Normal Mode Analysis (NMA), allows the grouping of the protein motions based on their frequency and hence can better identify the structural parts involved in concerted motion. In addition, the low-frequency motion modes revealed by this method are considered functionally relevant, e.g., as parts of enzyme catalysis, protein–ligand interactions, etc. (reviewed recently in [38]). NMA analysis was performed for the coarse-grained model of hPPA2, i.e., for the protein CA trace. Global low-frequency modes 1–3 calculated by this method are shown in Figure 8. The first two modes involve the motion of subunits with respect to each other: mode 1 (the slowest mode) is the rocking motion of subunits, mode 2 is the twisting motion of the subunits around the line connecting their centers. Modes 2 and 3 also involve the nodding and twisting motions of an Ω-loop 71–90. The description of conformational dynamics of the hPPA2 polypeptide backbone obtained by this method is in good agreement with the results of an all-atom PCA analysis of MD trajectories.

NMA analysis was additionally performed on the hPPA2 structures corresponding to several random frames of an MD trajectory, as well as on the average structure after equilibration and the original model before MD simulation; in all these structures, the global type of conformational changes was the same, which makes this result statistically significant (Appendix A).

Analogous NMA analysis was performed on the crystal structures of cytosolic PPases: human hPPA1, Sc-PPA1 and PPase from *Schistosoma japonicum* (PDB IDs 6C45, 1WGI, 2IHP and 4QLZ [12,27,28]) that share with hPPA2 the same subunit fold and dimeric organization but lack the Ω-loop. The results demonstrate that the rocking motion of the subunits described by mode 1 is very similar both in the two human PPases, hPPA1 and hPPA2 (Appendix A), and in other cytosolic PPases (Appendix A).

The oscillation of subunits found in dimeric PPases may be relevant to the information transfer between the two active sites of a dimer. In addition, this type of motion creates dynamic binding pockets between the subunits opening and closing and adopting various conformations in the course of PPase oscillation. Possible ligand binding in these pockets may impede the motion of subunits and fix one of the conformational states. Therefore, this observed type of motion may represent the basis for the allosteric regulation of PPase function by effectors binding at the subunit interface.

### 3.3. Effect of Pathogenic Mutations on Conformational Dynamics of hPPA2

Molecular dynamic simulation was performed on the models of mutant variants of hPPA2 carrying the mutations Ser61Phe, Met94Val, Met106Ile, and Gln294Pro analogous to the pathogenic natural variants. All these proteins were modeled in holoform analogous to the wild-type hPPA2 (Appendix A). Analysis of their structures obtained after equilibration demonstrates that these proteins retain the global structure typical for the WT hPPA2. However, their conformational dynamics demonstrate several differences compared to the WT enzyme. First, some mutations cause a change in the global flexibility of the protein. Appendix A shows the RMSF profiles of the mutant variants. Statistical analysis demonstrates that the variant Met94Val has a negative median value ΔRMSF of −0.52 Å compared to the WT PPase, meaning that this protein is more rigid than the WT enzyme. Other mutant variants have positive values of median ΔRMSF of 1.15–1.65 Å compared to the WT PPase, which means that their structures are overall more flexible.

Flexibility of the overall structures was also characterized by the analysis of the total amount of H-bonds in the dimeric proteins over the time course after equilibration. The results show that the Met94Val variant has on average more H-bonds than the WT hPPA2 (138 against 134), Gln294Pro variant—significantly fewer H-bonds (121 against 134), and the variants Ser61Phe and Met106Ile have a similar or slightly decreased count of H-bonds compared to the WT protein (131 and 132, respectively). These data support the increased flexibility of the Gln294Pro variant and decreased flexibility of Met94Val.

The results of a PCA analysis of the MD trajectories of the mutant variants are in good agreement with this conclusion. The values of PC1 (Table 1) for Ser61Phe, Gln294Pro and Met106Ile, as well as PC2 for the first two variants, are higher than for the WT hPPA2, which means that the protein motion along these principal components is more pronounced in the mutants than in the WT enzyme. For the Met94Val variant, the values of PC1 and PC2 are similar to the WT hPPA2, meaning that the overall conformational dynamics for this mutant are of a similar degree. For the Met106Ile variant, the motion along PC2 is less pronounced.

The plots of PC2 against PC1 for the mutant variants are shown in Appendix A. It can be seen that for Gln294Pro and especially Ser61Phe, the plots have higher dispersion around the most populated cluster, which suggests that these proteins are less ordered than the WT PPase. For Gln294Pro, the character of the plot suggests the increased flexibility of some local structural element rather than the overall structure. Analysis of the RMSF profiles (Appendix A) suggests that these regions with larger fluctuations may be region II (residues 152–164) and the chain fragment 280–320. The Met94Val and Met106Ile variants have compact and dense major clusters, and in Met94Val, it is disconnected from the rest of the points, which can be interpreted as the more rigid structure with more difficult transitions between the available conformational states. Ser61Phe also forms two different clusters, but the intermediate points between them are more populated than in Met94Val. In Met106Ile, the intermediate states are as populated as in the WT PPase, which suggests that the transitions between conformational states are not impaired in this variant.

The impact of mutations on the regions of the protein molecule that are important for the active site dynamics in the course of catalysis can be seen in Appendix A. Mutation Met106Ile decreases the RMSF of region I almost twofold compared to the WT PPase. Mutation Met94Val affects both regions I and II, though to a lesser degree. In contrast, mutation Ser61Phe makes regions I and III more flexible. In the variant Gln294Pro, the RMSF of the region II is increased by 20%.

Positions of the metal ions in the mutant variants Ser61Phe, Met94Val and Met106Ile are virtually the same as in the WT PPase, considering their self-fluctuations; in contrast, in the Gln294Pro variant, the M1 metal ion in subunit B oscillates between the two positions—one of which is 5.6 Å apart from the conventional binding site (Figure 9 and Appendix A). In the dimer of Gln294Pro superimposed to WT hPPA2, one subunit significantly deviates from its position of the WT protein as can be seen, for instance, from the different positions of the α-helices 261–279 (Figure 9A). The conformation of a protein chain region II (residues 152–164) is also different in Gln294Pro, where it adopts even more open conformation than in the WT protein, in addition to its larger RMSF values (Appendix A); this difference may cause the observed variations in the M1 position.

Another difference observed in the dynamics of the mutant variants is the change in the character and density of non-covalent interactions of the regions of protein in the vicinity of the mutation site. Figure 10 shows heatmaps of the contact density of the mutated residues in the structures of WT PPase and mutant variants calculated as the number of atoms that can be found within 5 Å of mutated residue over the MD trajectories. It can be seen that for all mutant PPases, mutation causes drastic changes in the observed heatmaps. For example, Ser61 in the WT hPPA2 has connections with residues 60–65, 114, 292, and the number of these connections is significantly decreased after mutation. On the other hand, the total number of diffused connections with the region 291–293 and in particular with the residue Thr291 is increased. In turn, the Gln294Pro variant as a result of mutation loosens connections with the residues 54–61, though the possible impact of this on the PPase function is unclear. In the mutant variant Met106Val, the connection density of mutated residue is significantly increased; this includes the regions 97–98, 104–107 and 117–119.

In the case of Met94Val, the general picture is not as clearly demonstrated; however, connections are lost with the regions 142–143 and 96–97, which can be important for the catalytic performance of PPase, since both Glu97 and Tyr142 are the active site residues participating in the metal ion M2 and/or substrate binding. In contrast, residues 170 and 186 increase the density of connections in Met94Val, which could be related to the decreased flexibility of this mutant variant. Asp169 preceding Val170 in the sequence is the ligand of metal ion M1, while the neighboring residues Asp166 and Pro167 are important for the formation of the attacking nucleophile. The decreased flexibility of this structural element caused by mutation Met94Val, together with the decreased flexibility of active site loops II and III (Appendix A), may impair the movements required for this stage of catalysis, and lead to the catalytic incompetence of this mutant variant.

We performed an NMA analysis of the global motion of the mutant PPases and expected that it would not show significant differences from the WT hPPA2 since this approach was performed in the coarse-grained variant, which is usually insensitive to point mutations. In Ser61Phe and Met94Val, mode 1 is indeed the same type of motion as in the WT PPase, although in mutants, the motion is more intense and involves additional structural elements. Unexpectedly, significant variations of dynamics were observed for Met106Ile and Gln294Pro variants (Appendix A). In the Gln294Pro variant, mode 1 includes the intense motion of the Ω-loop, and in Met106Ile, mode 1 includes only Ω-loop, while the motion of subunits is involved in modes 2 and 3. These changes are subtle, but they can be taken into consideration when the functional properties of mutant variants are studied.

## 4. Discussion

The principal objective of our work is the characterization of mitochondrial human PPase, hPPA2, in an attempt to understand the molecular bases of the pathogenic effects of natural mutations leading to severe mitochondrial dysfunction. This particular study is aimed at the characterization of the conformational dynamics of hPPA2 in the solution and the effect of pathogenic mutations on the observed patterns. There is a number of crystal structures of Family I PPases, both from prokaryotes and eukaryotes, but the solution structure is not yet known, so our understanding of the dynamic behavior of PPases is based solely on the comparative analysis of crystal structures in various complexes. This analysis demonstrates that the slight variations in the static structure caused by single-residue mutations cannot completely explain the changes in the functional properties of PPases. Therefore, it is probably more informative to characterize the effects of mutations on the structure and function of PPases using the analysis of conformational dynamics. In order to fill this gap, we applied a number of computational approaches to describe the dynamic behavior of hPPA2 and four of its mutant variants corresponding to natural pathogenic variants.

The characterization of the global conformational dynamics of hPPA2 revealed two novel types of protein structural change that have not been described earlier on the basis of crystal structures of PPases. First, in hPPA2 and three other dimeric eukaryotic PPases, the most prominent structural change is the rocking motion of the subunits. As the low-frequency mode, this type of motion can be relevant for the global PPase function. This motion affects the architecture of the intersubunit interface and hence can shape interactions with possible ligands of various nature binding at the interface. Second, in hPPA2, the subunit type of motion is accompanied at various degrees by the motion of the Ω-loop (residues 71–90), which is unique for the animal mitochondrial PPases. This polar residue-rich loop forms a potential binding epitope at the protein surface; its motion can be involved in the interaction of PPA2 with allosteric effectors and/or recognition by protein partners. The characterization of these interactions and their structural bases is crucial for the understanding of hPPA2 metabolic function and its regulation, so it will be the focus of future studies.

Several novel findings of this work support the existing model of PPase function and enrich our knowledge of PPase catalysis. Our data confirm that the subunit asymmetry observed earlier in several crystal structures of oligomeric PPases is a genuine result of protein dynamics rather than the artifact of crystal packing and suggests its importance for the catalysis. Results of this work show that, while both subunits are constantly in the “open” form, one subunit in the course of dynamics becomes more “open” than the other, and as a result has weaker contact with the metal ion M1. Since the shift in the conformational status and the motion of M1 are the important parts of PPase catalysis, the observed difference between subunits reflects their different progress along the catalytic cycle. A similar result has been observed earlier, for instance, in the structure of ScPPA1 2IHP [11], where subunits A and B correspond to different catalytic intermediates. Our data reveal that the subunits of a dimeric PPase can adopt the distinct conformational substates even within the single established catalytic intermediate (holoform) and allow a smoother description of a catalytic pathway.

This result is corroborated by another novel finding relevant to PPase catalysis: our data reveal the spontaneous formation of the binding site for the anionic ligands in the exact position where pyrophosphate is bound in the enzyme–substrate complex. This site was absent in the original structure but formed in the course of molecular dynamics, in particular, as a result of metal dynamics at site M1. This site was found in one subunit of a dimer, which represents more “open” conformation and has weaker contact with the M1 metal ion. Taken together, our results suggest that the observed conformational changes in this subunit may be a prerequisite for further substrate binding. This property may explain the homotropic cooperativity observed earlier for other oligomeric PPases, when one subunit of the oligomer “senses” the ligand binding in the active site(s) of the other subunit(s). Further research is required to explore the pathways of information transfer between the active sites.

Pathogenic effects of hPPA2 mutations on mitochondrial function result from the decrease in total PPase activity, which at the molecular level can be caused by the impaired protein fold, stability, or functional properties (including catalytic activity, regulation by effectors, and interactions with protein partners). Our work included the characterization of structural properties of mutant variants, which allowed some insight into possible functional implications.

Our data suggest that the pathogenic mutations are not expected to significantly impair protein folding or thermostability, so these reasons cannot be suggested as the primary cause of a general PPase incompetence. This prediction is in agreement with the previously characterized mutant variant Met52Val of OpPPA2 corresponding to the Met94Val variant of hPPA2 which had the same thermostability as the WT enzyme [11].

In support of that conclusion, the characterization of the conformational dynamics of the mutant PPases demonstrated that they retain the overall structure typical for the WT hPPA2, although with slight variations, and that this structure is stable over the course of MD simulation. Variations in the conformational stability were observed for the variants Ser61Phe and Gln294Pro that demonstrated larger fluctuations around the equilibrium state compared to the WT hPPA2. In addition, the Glu294Pro variant had a fewer H-bond stabilizing protein structure, which could imply lower thermo- and conformational stability. On the other hand, even slight changes in the global fold or the arrangement of surface structural elements can affect the interaction of PPases with cell partners, including proteases, so the predicted moderate changes in ΔΔG can affect the proteolytic stability of mutant variants or their recognition by cell degrading systems. Therefore, the stability of mutant PPases in the cell cannot be reliably predicted, and the decreased stability as the cause of mitochondrial dysfunction cannot be completely ruled out.

Another possible reason for the pathogenic effect of these mutations is their catalytic incompetence as PPases. The residues mutated in the variants studied in this work are not located in the active site and do not have an obvious connection to the PPase functional sites. However, our results suggest that these variants, similarly to other pathogenic variants of hPPA2 [8,9,10], can be less active than the WT enzyme, or even completely inactive. This suggestion is based on (a) the impairment of conformational dynamics of catalytically important structural elements found in all four mutants; (b) changes in the metal cofactor coordination found in Gln294Pro; and (c) changes in the connections of the catalytically important residues found in Met94Val. In terms of functional activity, these features observed from dynamics may be interpreted as the impaired cofactor or substrate binding and decreased catalytic power.

Met94Val is an example of a variant where the catalytically important regions are less flexible; in particular, the active site loop II is too strictly fixed due to the additional connections between the residues of this region and the mutated residue. We may suggest that, as a result, conformational changes required for the formation of the attacking nucleophile can be negatively affected in this mutant, which would significantly decrease its catalytic activity, as it was found for the Met52Val variant of OpPPA2 [11]. In contrast, the catalytically important residues Tyr142 and Glu97 that require firm anchoring in the hydrophobic core become less ordered with this mutation. As a result, the catalytic activity of Met94Val can be further decreased due to the impaired affinity for the cofactor or substrate.

Ser61Phe and Gln294Pro variants demonstrate enhanced flexibility up to the disordered character of motion, which can impair the pre-organization of the active site required for effective catalysis. In Gln294Pro, some regions are overly flexible, including region II, which determines the correct functioning of the system regulating the transition between the “open” and “closed” conformations of the active site. In this mutant, the system malfunctions and the M1 metal ion is positioned incorrectly. Therefore, impaired cofactor binding can be predicted for this variant that, in turn, would affect all consecutive stages of catalysis. In addition, less compact Ser61Phe and Gln294Pro proteins can be more prone to proteolytic degradation and/or aggregation.

The Met106Ile variant demonstrates decreased flexibility of some regions (e.g., loop I) and of the overall protein, and as a result, in the global protein dynamics, the motion of surface elements (e.g., the Ω-loop) prevails. Since the residues of loop I are involved in PP_i_ and P_i_ binding, these data may suggest the decreased affinity for these ligands. In support of this conclusion, an anionic binding site was not found in the structure of Met106Ile after MD simulation. Over-flexibility of the surface elements may also affect the protein recognition by the possible protein partners required, e.g., for the protein trafficking into the mitochondria, which can also be negatively affected by this mutation.

In conclusion, our results provide novel information on PPase catalysis in general and the effects of pathogenic mutations on hPPA2 in particular. A combination of factors changing the inherent patterns of protein dynamics can contribute to the overall failure of mutant PPases studied here to maintain the necessary mitochondrial function. They include the impaired conformational transitions of functionally important elements leading to catalytic deficiency, improper positioning of functional ligands exemplified here by cofactor metal ions, and probably changes in protein recognition by other cell components. This work provides a basis for the further study of PPase dynamics and the understanding of molecular mechanisms of PPase function in normal and pathological circumstances. Future research will focus on the characterization of the hPPA2 metabolic role and regulation, including a search of its cell effectors and protein partners, and molecular dynamics of its functionally relevant complexes.

## Figures and Tables

**Figure 1 life-15-00100-f001:**
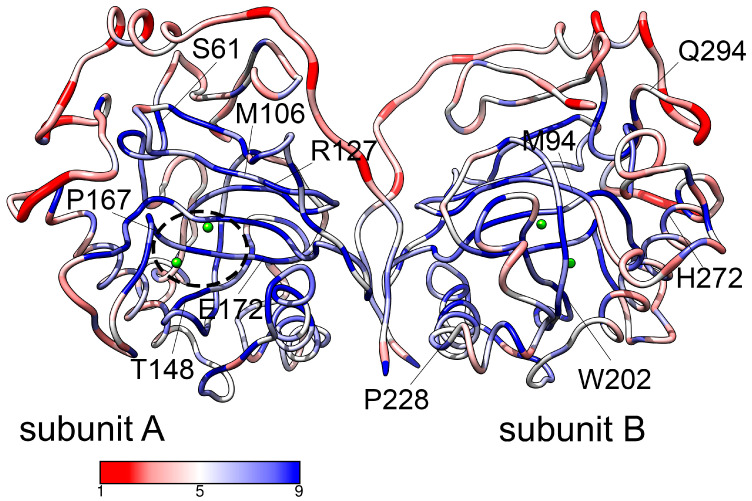
The model of hPPA2 dimer (composed of subunits A and B) colored by conservation grade assigned by ConSurf [23,24]. The color key is shown below the structure, grade 1 being the lowest conservation and grade 9 the highest conservation. Line thickness corresponds to the grade (9, thinner; 1, thicker). Positions of the residues whose mutations are pathogenic are labeled. The active site position is marked in the subunit A by a dashed ellipse. Mg^2+^ ions are depicted as green spheres.

**Figure 2 life-15-00100-f002:**
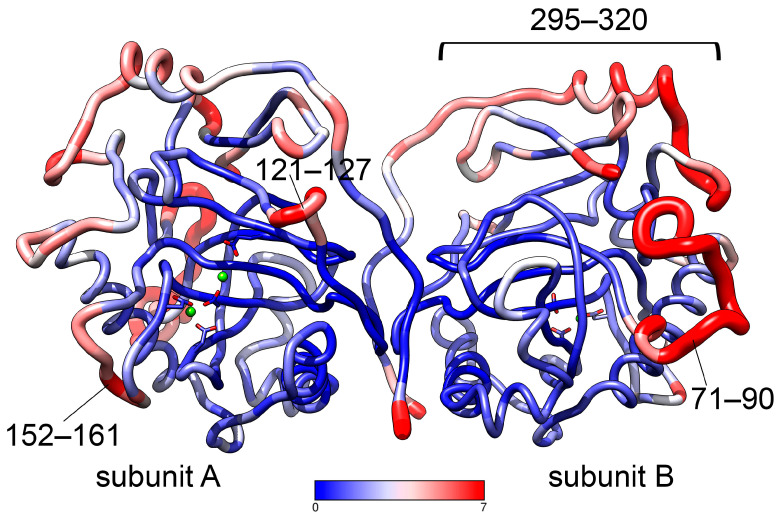
Dimer of hPPA2 (average structure after equilibration) colored by all-atom RMSF. The color key corresponding to RMSF values is shown below the structure. Metal ions are shown as green spheres, and their protein ligands are depicted in stick representation.

**Figure 3 life-15-00100-f003:**
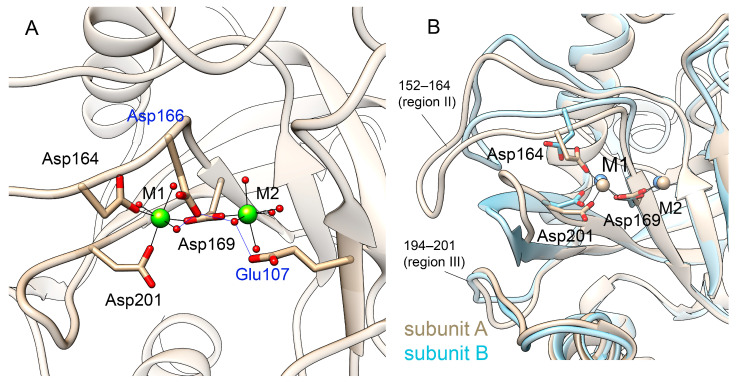
Mg^2+^ binding in the hPPA2 model. (**A**) Coordination of Mg^2+^ in the representative structure of the simulation trajectory. Mg^2+^ ions are shown as green spheres, first shell coordination is shown as black lines and second shell coordination with protein residues is shown in blue lines. Protein ligands of metal ions are labeled in black and the second coordination shell ligands are labeled in blue. (**B**) Coordination of Mg^2+^ in subunits A (shown in tan) and B (shown in blue) of dimeric hPPA2. Metal ions are the same color as the protein.

**Figure 4 life-15-00100-f004:**
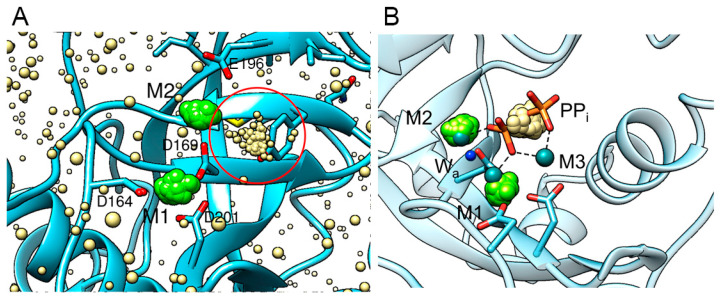
Anionic binding site in hPPA2. (**A**) A distribution of Cl^-^ in the simulation box; clustering of Cl^-^ in one of the subunits is shown in a red circle. Random frames from MD simulation are superimposed by backbone atoms of a protein; protein molecules from all but one frame are omitted for clarity. Mg^2+^ ions at M1 and M2 sites are shown as green spheres and Cl^-^ as smaller yellow spheres. (**B**) Close-up view of the active site with all Cl^-^ ions except for those in the cluster omitted for clarity. The crystal structure of the enzyme–substrate complex of homologous PPase from *E. coli* (Ec-PPase, PDB ID: 2AUU [14]) is superimposed to the structure of hPPA2; protein ribbon is only shown for hPPA2 while the ligands of an enzyme–substrate complex are shown from the Ec-PPase. Pyrophosphate PP_i_ is shown as orange/red sticks and Mg^2+^ ions from the enzyme–substrate complex are shown as blue-green spheres. Small blue sphere marked as W_a_ is an attacking nucleophile replaced by F^-^ in the crystal structure.

**Figure 5 life-15-00100-f005:**
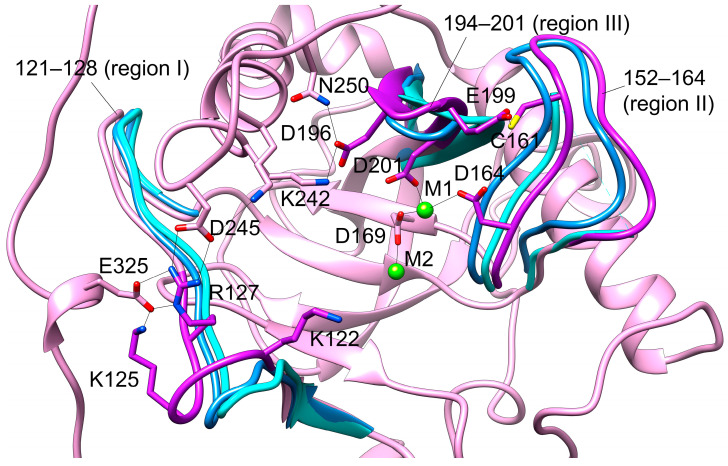
The active site of a representative conformation of hPPA2 (pink) compared to the “open” and “closed” conformations of Sc-PPase (blue and cyan, PDB ID: 2IHP [12]). Flexible regions I, II and III of hPPA2 are highlighted in purple. Corresponding regions of Sc-PPase are shown in blue (“open” state) and cyan (“closed” state). The residues of hPPA2 coordinating metal ions as well as several other active site residues discussed in the text are depicted in sticks. Mg^2+^ ions are shown as green spheres. Protein-metal ions coordination, H-bonds and salt bridges between residues are shown as black lines.

**Figure 6 life-15-00100-f006:**
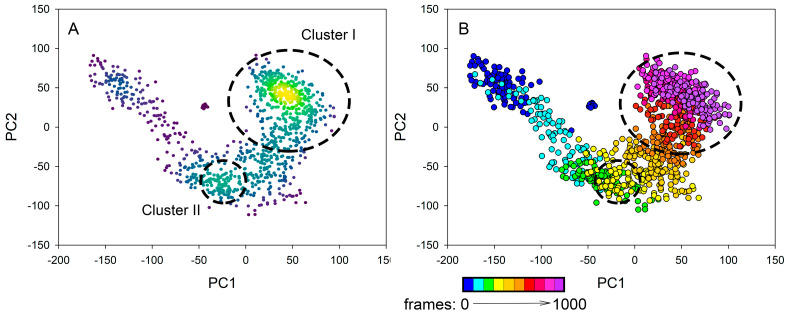
PCA analysis of the MD trajectory of WT hPPA2. Conformational space of a protein molecule is illustrated by the two first principal components PC2 vs. PC1. (**A**) Data are colored by population density using a temperature scale from purple (lowest density) to yellow (highest density). (**B**) The same data colored by a frame range from blue (first 100 frames) to purple (last 100 frames), as depicted by the color key below the plot.

**Figure 7 life-15-00100-f007:**
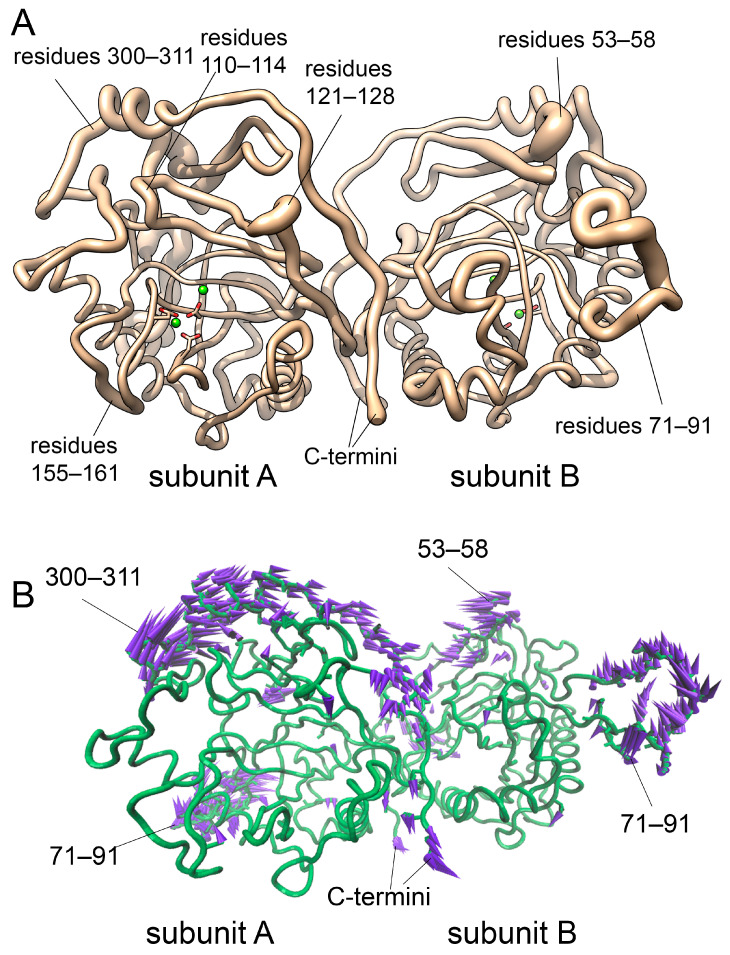
(**A**) Contribution of hPPA2 residues to PC1 as depicted by the thickness of a tube. Position of the active site is shown by the bound Mg^2+^ ions (green spheres). (**B**) Direction of protein atoms movement in PC1 (shown by the purple arrows; arrow length is proportional to the atom fluctuation).

**Figure 8 life-15-00100-f008:**
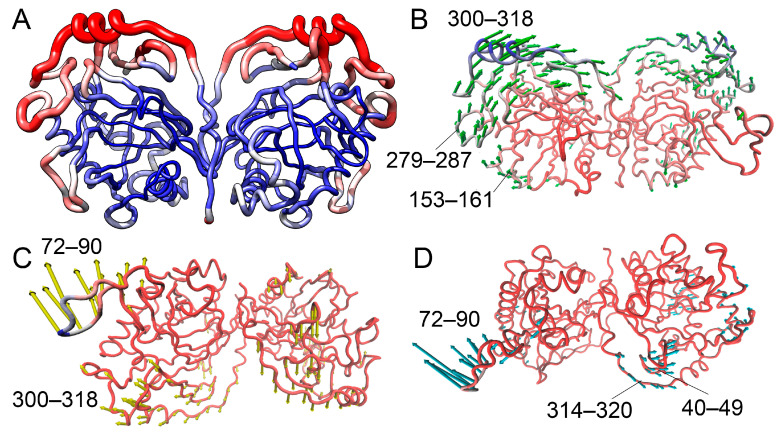
Global modes 1–3 of hPPA2 motion calculated according to the coarse-grained Normal Mode Analysis using DynOmics server [34]. (**A**) Dimer of hPPA2 colored according to the contribution of the residues into mode 1 (larger contribution corresponds to red color and thicker tube). (**B**–**D**) Porcupine plots showing directions of residue motion in modes 1–3: (**B**) mode 1; (**C**) mode 2; (**D**) mode 3. Directions of the C_α_ atom motion are shown by the arrows of different colors. Arrow length is proportional to the atom fluctuation. Fluctuations less than 2.0 Å are omitted for clarity. Residue numbers are labeled for the chain fragments involved in the motion.

**Figure 9 life-15-00100-f009:**
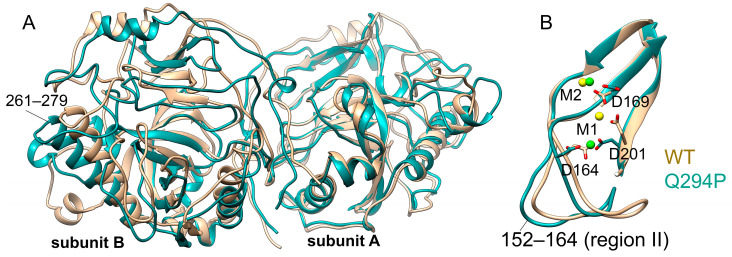
Superposition of Gln294Pro (dark cyan) and WT hPPA2 (tan). Representative structures after equilibration are taken for both PPases. (**A**) Overall structures of dimeric PPases superposed by subunits A. (**B**) Close-up view of metal binding sites in subunit B of the two proteins. Mg^2+^ ions are shown as spheres (green in Gln294Pro, yellow in WT hPPA2), and their ligands are shown in stick representations.

**Figure 10 life-15-00100-f010:**
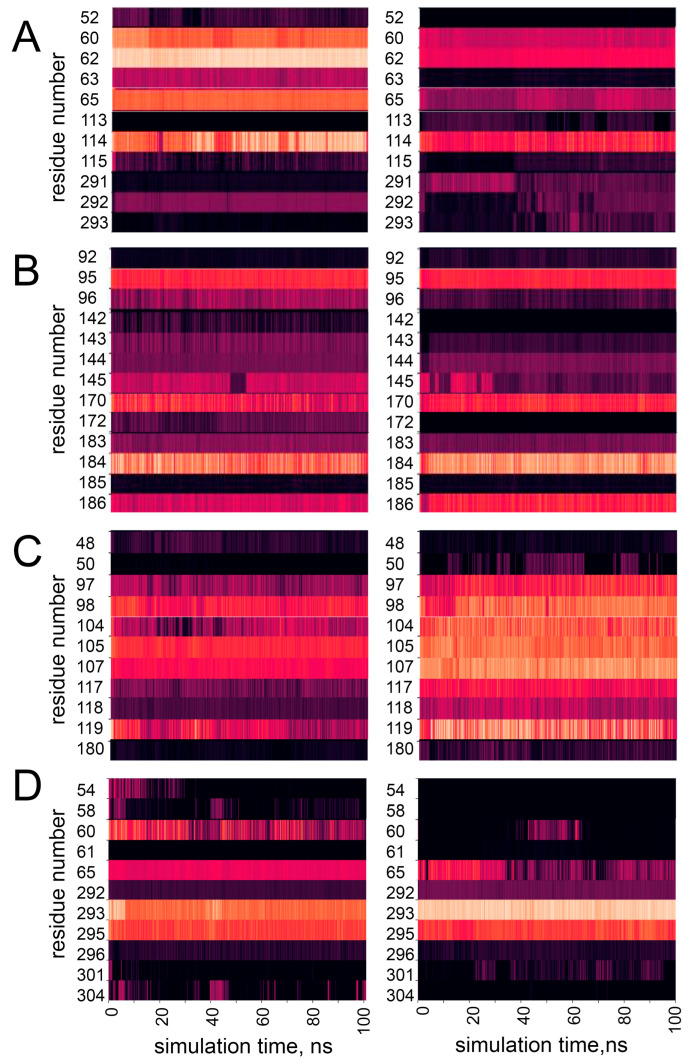
Dynamics of the contacts of mutated residues in the structures of WT hPPA2 and the mutant variants over the MD trajectories. (**A**) Ser61Phe, (**B**) Met94Val, (**C**) Met106Ile, (**D**) Gln294Pro. X axis is the simulation time; Y axis is the selected residue numbers along the chain. Each heatmap shows the density of contacts, i.e., how often the atoms of these residues can be found within 5 Å of the mutated residue in the WT PPase (**left** pictures) or after mutation (**right** pictures). Darker color corresponds to lower density of contacts, brighter color—to the higher density of contacts.

**Table 1 life-15-00100-t001:** The mean values of the first two principal components, PC1 and PC2, for WT hPPA2 and its mutant variants.

Protein	PC1, Å2	PC2, Å2
WT hPPA2	4232	2431
Ser61Phe	7709	2753
Met94Val	4419	2025
Met106Ile	7241	1738
Gln294Pro	6163	3397

## Data Availability

The data are available on request from the corresponding author.

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
