# Peer review of "Conformational Dynamics of Mitochondrial Inorganic Pyrophosphatase hPPA2 and Its Changes Caused by Pathogenic Mutations"

_life, 2025, doi:10.3390/life15010100_

Round 1

Reviewer 1 Report

Comments and Suggestions for Authors

Bezpalaya E.  et all  discovered a deeper 22 insight into the structural basis of PPase function and the possible effects of pathogenic mutations 23 on the protein structure and function.  I like the manuscript but I would like to add more info 

1) Author needs to validated 3D molecular structure in cells, at least one experiment that can confirm and predict investigations

2) The discussion is very long and hard to understand, I suggest them rewrite discussion

Reviewer 2 Report

Comments and Suggestions for Authors

This is a high quality manuscript describing the conformational dynamics of mitochondrial inorganic pyrophosphatase hPPA2 and its changes caused by pathogenic mutations. Authors described the wild-type enzyme and four mutant variants of hPPA2 which were related to the natural pathogenic variants causing severe mitochondrial disfunction and cardio pathologies. Authors identified the global type of flexible motion that was shared by other dimeric PPases. This motion is discussed in terms of allosteric behavior of the protein. Authors suggest molecular mechanisms of the functional incompetence of hPPA2 caused by mutations. The results of this work provide a novel insight into the structural basis of PPase function and the potential pathogenic mutations on the protein structure and function.

Author Response

Thank you very much for taking the time to review this manuscript.

Reviewer 3 Report

Comments and Suggestions for Authors

Introduction

    • The introduction covers too many topics, making it difficult to discern the main focus of the study.
    • The aim of the study is introduced at the end (lines 86–94), making it difficult for the reader to understand the purpose early on.
    • The relationship between the mentioned background information and the specific study objectives is not well-established.
    • Important background concepts, such as the difference between hPPA1 and hPPA2 or their relevance to human disease, are introduced in a dense and fragmented manner.
    • Although the introduction mentions poorly understood regulatory mechanisms, it does not clearly articulate the specific gaps in knowledge that this study addresses.
    • The significance of understanding conformational dynamics in hPPA2 is not strongly highlighted.
    • Certain points, such as the roles of methionine residues or pathogenic variants, are mentioned multiple times, which can confuse readers.

                          Discussion

    • The discussion meanders between various points, making it hard to follow the main conclusions and their implications.
    • It jumps between specific findings (e.g., effects of specific mutations) and broader observations (e.g., general dynamics of PPases) without clear transitions or structure.
    • Several points from the results section are reiterated verbatim without further interpretation or broader context, reducing the impact of the discussion.
    • Redundant statements about known PPase properties and their dynamics dilute the focus on novel findings.
    • While the discussion highlights specific findings, it does not sufficiently explain their broader implications for understanding PPase function, mitochondrial dysfunction, or therapeutic potential.
    • The relevance of findings to previous studies or their novelty is not adequately emphasized.
    • The relationship between stability, dynamics, and catalytic efficiency of mutant PPases is not tied to the broader discussion on pathogenicity.
    • Potential applications of the findings, such as drug development or clinical implications, are not explored.
    • The conclusion section reiterates points without summarizing the key takeaways or highlighting directions for future research.
    • It lacks a compelling statement of the study’s impact.

                    Results

    • The text is dense with technical information, making it hard to follow for readers unfamiliar with molecular dynamics and PPase-specific terminologies.
    • Overly detailed descriptions (e.g., specific positions of residues and interactions) distract from the broader significance of the findings.
    • Key findings are buried in lengthy paragraphs, reducing their impact.
    • The section focuses heavily on computational details without adequately linking findings to the biological or clinical implications, such as mitochondrial dysfunction or disease.
    • The broader impact of the mutations (e.g., Met94Val’s role in catalytic activity) is underexplored.
    • Repeated mention of concepts like subunit asymmetry and motion dynamics without offering new insights.
    • Overemphasis on the methodological details that could be streamlined.
    • Unique discoveries, such as the impact of Met94Val on active site flexibility or the spontaneous formation of an anionic binding site, are not sufficiently highlighted.
    • The importance of the observed W-loop motion is not clearly emphasized.
    • The effects of different mutations on dynamics, stability, and catalytic properties are presented in isolation without synthesis.

o   Comparisons across mutants are limited and lack interpretive depth.
